

# Isolation and characterization of a motility-defective mutant of *Euglena gracilis*

Shuki Muramatsu[1,2], Kohei Atsuji[2,3], Koji Yamada[2,3], Kazunari Ozasa[4], Hideyuki Suzuki[2], Takuto Takeuchi[2], Yuka Hashimoto-Marukawa[2,3], Yusuke Kazama[5,6], Tomoko Abe[5], Kengo Suzuki[2,3] and Osamu Iwata[2]

[1] Department of Health Science, Showa Women's University, Tokyo, Japan
[2] euglena Co., Ltd., Tokyo, Japan
[3] Baton Zone Program, RIKEN, Saitama, Japan
[4] Bioengineering Laboratory, Cluster for Pioneering Research, RIKEN, Saitama, Japan
[5] RIKEN Nishina Center, Saitama, Japan
[6] Faculty of Bioscience and Biotechnology, Fukui Prefectural University, Fukui, Japan

## ABSTRACT

*Euglena gracilis* is a green photosynthetic microalga that swims using its flagellum. This species has been used as a model organism for over half a century to study its metabolism and the mechanisms of its behavior. The development of mass-cultivation technology has led to *E. gracilis* application as a feedstock in various products such as foods. Therefore, breeding of *E. gracilis* has been attempted to improve the productivity of this feedstock for potential industrial applications. For this purpose, a characteristic that preserves the microalgal energy e.g., reduces motility, should be added to the cultivars. The objective of this study was to verify our hypothesis that *E. gracilis* locomotion-defective mutants are suitable for industrial applications because they save the energy required for locomotion. To test this hypothesis, we screened for *E. gracilis* mutants from Fe-ion-irradiated cell suspensions and established a mutant strain, $M_3^-$ZFeL, which shows defects in flagellum formation and locomotion. The mutant strain exhibits a growth rate comparable to that of the wild type when cultured under autotrophic conditions, but had a slightly slower growth under heterotrophic conditions. It also stores 1.6 times the amount of paramylon, a crystal of $\beta$-1,3-glucan, under autotrophic culture conditions, and shows a faster sedimentation compared with that of the wild type, because of the deficiency in mobility and probably the high amount of paramylon accumulation. Such characteristics make *E. gracilis* mutant cells suitable for cost-effective mass cultivation and harvesting.

# INTRODUCTION

The locomotion of organisms is directly linked to characteristic evolutionary survival competitions, such as acquiring food, finding suitable environments, and escaping from predators (*Domenici, Claireaux & McKenzie, 2007*). Locomotion in response to environmental stimuli is defined as "taxis", including chemotaxis, gravitaxis, and

Corresponding authors
Koji Yamada, yamada@euglena.jp
Kengo Suzuki, suzuki@euglena.jp

phototaxis, which respectively refer to locomotion toward chemicals, gravity, and light, respectively (*Dusenbery, 2009*). Such taxes are essential for organisms to survive in natural environments and are regulated by complex mechanisms (*Webre, Wolanin & Stock, 2003*; *Okita, 2005*; *Roberts, 2006*; *Jékely et al., 2008*; *Roberts, 2010*; *Sourjik & Wingreen, 2012*). The means of locomotion depend on the organism size and are optimized through evolution (*Jahn & Bovee, 1965*; *Dusenbery, 1997*). Many of the motile unicellular microorganisms in the hydrosphere use their flagella to swim (*Miyata et al., 2020*).

Among the microalgae, *Chlamydomonas reinhardtii* has been widely used as a model organism to study the molecular mechanisms underlying the swimming of phytoplankton cells using their flagella. *C. reinhardtii* is used as a model microalga because its genetic methodology has been established (*Harris, 2001*). Using their gravitaxis and phototaxis, strategies to screen non-motile *C. reinhardtii* mutants were developed (*Kamiya, 1991*). The acquisition of these mutants and their analysis elucidated the molecular mechanisms used for swimming; e.g., the interaction between microtubule and dynein motor protein drives the motion of the flagella (*Kamiya & Okamoto, 1985*; *Kamiya, Kurimoto & Muto, 1991*; *Kamiya, 1995*). The photosynthetic flagellate species of the genus *Euglena* also use their flagella to swim, but they can be shed in response to chemical or mechanical stimuli (*Bovee, 1982*). In contrast to *Chlamydomonas* cells, which cannot change their shape, *Euglena* spp. demonstrate an amoeboid movement called "euglenoid movement" without a flagellum, although it is an extremely slow means of migration (*Bovee, 1982*).

*Euglena gracilis* is extensively used as a model organism to study the mechanisms of photosynthesis, cell metabolism, and locomotive behaviors, such as gravitaxis and phototaxis (*Richter et al., 2002*; *Daiker et al., 2010*; *Daiker et al., 2011*; *Schwartzbach & Shigeoka, 2017*). Recent advances in experimental techniques, such as gene knockdown, have facilitated *E. gracilis* gene function studies. For example, a photoactivated adenylyl cyclase was reported to be required as a photoreceptor for phototaxis (*Iseki et al., 2002*). In contrast, it remains very difficult to obtain nuclear mutants of *E. gracilis*; few traits are induced by mutagens, such as ultraviolet light or ethyl methanesulfonate, which has hampered the complete inhibition of a specific gene functions, and is suggested to be related to the species' polyploidy (*Schiff & Epstein, 1965*; *Hill, Schiff & Epstein, 1966*). Accordingly, few locomotion-defective mutant strains have been identified and characterized (*Schiff, Lyman & Russell, 1980*). Although a chloroplast mutant of *E. gracilis*, $M_2^-$BUL, was previously reported to lack motility (*Shneyour & Avron, 1975*), it has not been extensively examined; it is reported to lack photosynthesis activity and is not available in current culture collections, e.g., The Sammlung von Algenkulturen der Universität Göttingen (Culture Collection of Algae at Göttingen University [SAG], DEU), to which various *E. gracilis* wild-type strains are deposited.

In addition to its utilization as a model organism, *E. gracilis* has been exploited for industrial applications because of its fast proliferation, nutrient-rich features, and characteristic metabolism (*Schwartzbach & Shigeoka, 2017*). As a saccharide reserve, *E. gracilis* stores paramylon, a crystalline form of $\beta$-(1,3)-D-glucan, which has various applications in the food industry (*Watanabe et al., 2013*; *Russo et al., 2017*). Thus, *E. gracilis* cells are particularly suitable as an ingredient in functional foods and supplements.

Moreover, the stored paramylon is metabolized under hypoxic conditions into wax ester, which is suitable as a biofuel source (*Inui et al., 1982*; *Inui et al., 1983*). To enhance these industrial applications, *E. gracilis* has been bred to achieve thermostability and high oil production using Fe-ion irradiation as a mutagen (*Yamada et al., 2016a*; *Yamada et al., 2016b*).

In the present study, we produced and characterized a non-motile mutant strain of *E. gracilis* ($M_3^-$ZFeL). We hypothesized that the non-motile mutant strain has a higher paramylon production than the wild type, because it could save significant energy, which is mostly used for the swimming in the wild type (*Hamilton, Nakamura & Roncari, 1992*; *Tunçay et al., 2013*). We used our previously established method, which uses Fe-ion beam irradiation as a mutagen agent, to produce non-motile mutants of *E. gracilis* (*Yamada et al., 2016a*; *Yamada et al., 2016b*). We screened for non-motile cells from the Fe-ion beam-irradiated population and selected $M_3^-$ZFeL for further studies. The proliferation rate, swimming and sedimentation speed, and paramylon and wax ester production of $M_3^-$ZFeL were measured. Our results demonstrated that Fe-ion beam irradiation and the following screening method was an efficient strategy necessary to acquire non-motile *E. gracilis* mutant strains, as well as the potential of the produced $M_3^-$ZFeL strain for industrial application.

## MATERIALS & METHODS

### Strains, culture, and media

The wild-type Z strain, which is identical to *E. gracilis* SAG 1224-5/25, was provided by the culture collection of the Institute of Applied Microbiology (IAM), University of Tokyo, Tokyo, JPN. The $M_3^-$ZFeL mutant strain produced in this study is deposited in the microbial culture collection at the National Institute for Environmental Studies (NIES, Tsukuba, JPN) with registration number NIES-4440. The cells were cultured using Cramer–Myers (CM) (*Cramer & Myers, 1952*) and Koren–Hunter (KH) media (*Koren & Hutner, 1967*) for autotrophic and heterotrophic growth, respectively, prepared at pH 3.5. The CM medium does not include a carbon source, while the KH medium includes 12 g $L^{-1}$ of glucose and various organic acids and amino acids as carbon sources. The seed culture was maintained in KH medium with each month of subculture.

### Fe-ion irradiation and mutant screening

The mutant strain was obtained through Fe-ion irradiation using the same conditions as previously reported (*Yamada et al., 2016a*; *Yamada et al., 2016b*), and mutant screening was based on the algal phenotypes. Two milliliters of an *E. gracilis* cell suspension ($4 \times 10^5$ cells $mL^{-1}$) was placed in hybridization bags that were segmented into $5 \times 7$ cm compartments. They were then irradiated by Fe ions (linear energy transfer [LET]: 650 keV $\mu m^{-1}$) at a dose of 50 Gy in the RIKEN RI-beam factory (Wako, Saitama, JPN). After one week of recovery by culture inoculation in KH culture medium, two rounds of mutant screening were conducted.

About $1 \times 10^5$ mutagenized cells with independent genetic backgrounds were placed at one end of a culture dish filled with 10 mL of liquid KH medium with 0.5% methyl

cellulose, which was added to solidify and prevent unexpected stirring of the culture, and were illuminated from the opposite end with 50 μmol photons $m^{-2} s^{-1}$ of fluorescent light. After one week of culture incubation following Fe-ion beam irradiation, cells proliferated more than 100 times, resulting in $1 \times 10^7$ mutagenized cells, and most of them were found at the illuminated side through phototaxis. About 1,000 of the cells remained at the initial inoculation point; these were collected as putative non-motile mutant cells with a micropipette.

The collected population was supposed to include a high proportion of motile cells, which resided at the initial inoculation point due to random dispersal; therefore, they were subjected to a second round of screening. The cells were placed at the opening end of a 15 mL conical tube filled with 5 mL of liquid KH medium with 0.5% methyl cellulose, illuminated with 50 μmol photons $m^{-2} s^{-1}$ of fluorescent light, and shaded with aluminum foil, except at the closing end. After 2 weeks of static horizontal incubation in the tubes, the cells proliferated more than 1,000 times. Most cells were gathered at the unshaded end to seek light. We retrieved about 1,000 of the cells that showed no movement from the initial inoculation position and cultured them for an additional two weeks.

The proliferated cells were randomly isolated using fluorescence-activated cell sorting (MoFlo XDP; Beckman Coulter, Brea, CA, USA) to establish clonal lines in individual wells of 96-well plates (Tissue Culture Test Plate; TPP, Trasadingen, CHE) filled with 200 μL of KH medium. Populations without cell motility were selected by microscopic observations at 26 °C and identified as the motility-defective strains. The above-mentioned screening was performed for two independently mutagenized populations. During the screening process, cells proliferated more than $10^6$ times; therefore, each population included many genetically identical cells. Based on this possibility, one mutant strain was established from each population, i.e., two strains were established from the two independent populations.

## Morphological characterization

The cells were observed under an upright light microscope (DM2500B; Leica, Wetzlar, DEU) equipped with a differential interference contrast module. Scanning electron microscopy (SEM) was conducted using a field emission scanning electron microscope (SU8200; Hitachi, Tokyo, JPN) after following a general sample preparation. Briefly, 1 mL of culture in KH medium was centrifuged at $2,000 \times g$ for 1 min to collect the cells. The precipitated cells were pre-fixed by adding a solution containing 2.5% glutaraldehyde, 2% paraformaldehyde, and 0.1 M sodium cacodylate, and fixed again using a solution containing 1% osmium tetroxide and 0.1 M sodium cacodylate. The fixed samples were completely dried using a critical point dryer, sputter-coated with osmium, and then subjected to SEM photographing.

For quantifying the proportion of cells with flagella, the cells were cultured in CM or KH medium and used in their logarithmic growth phase for analysis. For the CM medium, the cells were cultured in 50 mL of medium using a 100 mL volume test tube aerated with 50 mL $min^{-1}$ of air containing 5% $CO_2$. For the KH medium, each strain was cultured in 50 mL of medium using a 100 mL volume conical flask with rotary shaking at 100 rpm. Each culture was conducted at 26 °C with 100 μmol $m^{-2} s^{-1}$ of constant illumination

and sub-cultured every week to maintain a stable proliferation. The cells were then fixed with glutaraldehyde (0.025%) and observed for the presence of flagella under an inverted microscope (CKX41; Olympus, Tokyo, JPN) equipped with a phase contrast module. To exclude subjective judgment, short and intact flagella were not differentiated and were counted as cells with flagella.

## Growth tests

The algae growth rate was evaluated in 100 mL test tubes containing 50 mL of medium. The cells were inoculated with an initial optical density (OD) of 0.1 and precultured for three days in 100 mL volume conical flasks containing KH medium with continuous shaking (100 rpm at 26 °C, and 100 m$^{-2}$ s$^{-1}$ of constant illumination). The cultured cells were then collected by centrifuging at 2,000 × g for 5 min and washed twice with either KH or CM medium, which was subsequently used for the culture test. Next, the cell suspension was inoculated in culture tubes with an initial OD of 0.1 and incubated at 26 °C with 100 µmol m$^{-2}$ s$^{-1}$ of constant illumination. KH and CM cultures were aerated with 50 mL min$^{-1}$ of air containing 5% $CO_2$. ODs from culture growth were determined over time by spectrophotometer (UVmini-1240; Shimadzu, Kyoto, JPN) at λ 680 nm, immediately after placing the suspension in the cuvette.

## Quantitative analysis of motility

We used microscopic observations with video image capture to quantitatively evaluate algae motility. Observation and image processing techniques were used as previously reported (*Ozasa et al., 2011*). We confined the cells in suspension (0.7 µL, containing 300–500 cells) in a closed circular polydimethylsiloxane (PDMS) micro-chamber (2.49 mm diameter and 140 µm depth), and observed red-light, bright-field transmission images with a 5 × objective lens at 26 °C. Approximately 300–500 cells were measured in the micro-chamber. Cell movements were visualized by differentiating, thresholding, and superimposing the video images sequentially (trace image), and evaluated by counting the spatial sum of the trace pixels in the trace image (*Ozasa et al., 2013*; *Ozasa et al., 2014*). The number of trace pixels was named the "trace momentum" (TM) (*Ozasa et al., 2013*; *Ozasa et al., 2014*). The measurement rate of the TM values was 0.67 Hz (one frame per 1.6 s). The TM value is a fair measure of all locomotive activity observed within the chamber. The typical swimming traces were individually superimposed onto the cell distribution image and prepared as a binary image to demonstrate the cell's swimming activity (*Ozasa et al., 2016*).

## Sedimentation analysis

Cellular sedimentation rates in KH and CM media at 26 °C were evaluated in 1.5 mL microtubes. The culture was prepared as indicated above for the morphology analysis. In brief, when CM medium was used, cells were cultured in 100 mL volume test tubes with aeration, whereas with KH medium, they were cultured in a 100 mL volume conical flask with rotary shaking at 100 rpm. Each culture was conducted with constant light illumination (100 µmol m$^{-2}$ s$^{-1}$). Culture ODs average at λ 680 nm were 23.4 and 22.0 by KH culture, and 4.1 and 4.4 by CM culture of wild-type and M$_3^-$ZFeL strains, respectively. Images of the cell suspension in each microtubule were taken at 1–5 min intervals with

a general compact digital camera. The transparent supernatant area was detected and quantified from the images using the Image J software, with the threshold values set as 0 to 50 after converting to greyscale. Identical rectangular regions were selected inside the images of microtube images, and the transparent area inside each region was quantified.

To evaluate the sedimentation speed at 20 cm depth, the *E. gracilis* wild type and the $M_3^-$ZFeL mutant strains were cultured in cuboid acryl beakers ($10 \times 10 \times 30$ cm) using 2 L of CM medium at 29 °C and 1,300 $\mu$mol m$^{-2}$ s$^{-1}$ of overhead illumination (light–dark ratio of 12:12 h). Beakers were filled with approximately the same concentrations (g L$^{-1}$) of cells, 0.37 and 0.32 g L$^{-1}$ for the wild-type and M$^-$3ZFeL strains, respectively. After stirring the culture, sedimentation was periodically observed for 3.5 h, and supernatant was carefully removed to obtain a concentrated culture. Sediment concentration was then quantified by weighing the dried cells after filtration with a glass fiber filter (GA-55; ADVANTEC, Tokyo, JPN).

## Quantification of carbohydrates and lipids

The harvested algal cells were freeze-dried (FDV-1200; EYELA, Tokyo, JPN), and the deproteinized paramylon and lipid components were extracted as previously reported (*Inui et al., 1982*; *Suzuki et al., 2015*). The dried cells (10 mg) were sonicated twice in 10 mL of acetone for 90 s (UD-201; TOMY, Tokyo, JPN) with dial setting at 4 in 50 mL volume of plastic centrifuge tubes. The paramylon was subsequently separated from the residual components using centrifugation, boiled for 30 min in 10 mL of 1% sodium dodecyl sulfate aqueous solution, and washed twice with 10 mL of water. The extracted paramylon was then quantified using the phenol–sulfuric acid method (*Montgomery, 1957*), which can quantify total carbohydrates. Similarly, neutral lipids were extracted from 100 mg of dried cells using n-hexane as a solvent; 10 mL of n-hexane was added to the dried cells in 50 mL glass centrifuge tubes. The suspension was then homogenized for 90 s using a sonicator (UD-201; TOMY, Tokyo, JPN) with dial setting at 4, and then filtered with a piece of glass fiber filter paper (GF/C; Whatman, Little Chalfont, Buckinghamshire, UK), followed by an additional step of residue extraction. After evaporating the collected organic solvent dissolving lipids, the weight of the residue left in the flask was quantified as that of the extracted total neutral lipid.

## Statistical analyses

The results with error bars, except that of motility quantification, are represented as mean $\pm$ standard error from three independent experiments. The results of motility quantification are represented as mean $\pm$ standard deviation (SD) for the whole measured time points. Statistical significance was analyzed using Student's *t*-test. For multiple comparisons, Bonferroni's corrections were applied. $p < 0.05$ was considered significant.

# RESULTS

## Characterization of $M_3^-$ ZFeL

Through our screening for non-motile mutants of *E. gracilis*, we established two mutants with independent genetic backgrounds. The mutant strains were named $M_3^-$ZFeL and
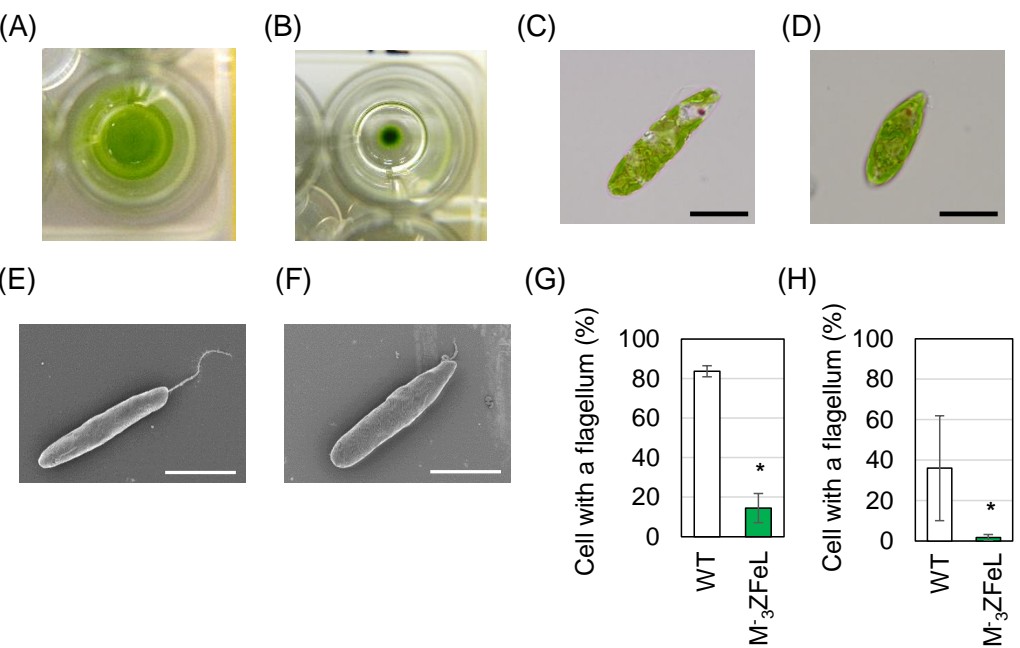

**Figure 1** $M_3^-$ZFeL and wild-type strains of *Euglena gracilis* in static culture. (A and B) Photographs of wild-type (A) and $M_3^-$ZFeL (B) colonies grown in Koren–Hunter (KH) medium on a 96-well plate for one week. (C–F) Microphotographs (C and D) and scanning electron micrographs (E and F) of wild-type (C and E) and $M_3^-$ZFeL (D and F) cells cultured in 100 mL volume conical flasks containing KH medium with continuous shaking (100 rpm, 26 °C, and 100 $\mu$mol m$^{-2}$ s$^{-1}$ of constant illumination). Scale bars indicate 20 $\mu$m. (G and H) The proportion of cells that possess flagella among the wild-type and $M_3^-$ZFeL cells in the Cramer–Myers (CM) (G) and KH media (H). More than 200 cells were observed for each condition in one day. The observations were conducted three times on different days. Error bars show the standard errors for three replicates. * $p < 0.05$, Student's *t*-test.

$M^-_4$ZFeL according to the typical nomenclature (*Schiff, Lyman & Russell, 1980*), with the phenotypic designation "M" for motility and the mutagen designation "Fe" for Fe-ion irradiation (*Yamada et al., 2016a*). The strain $M_3^-$ZFeL showed a stable behavioral phenotype, and was thus used in this study.

The *E. gracilis* $M_3^-$ZFeL strain showed non-motile phenotype with few defects in proliferation. This phenotype was identified by the formation of colonies from single-cell inoculation in liquid culture. The $M_3^-$ZFeL strain showed colony formation in liquid KH culture medium after a week, whereas wild-type cells were dispersed in the same culture medium (Figs. 1A and 1B). As judged by both light and electron microscopy, the $M^-_3$ZFeL cells were more rounded than were the wild-type cells. In addition, the $M_3^-$ZFeL flagella were shorter than those of the wild type (Figs. 1C–1F). Moreover, a significantly higher proportion of $M_3^-$ZFeL cells lacked flagella both in the CM ($p = 5.6 \times 10^{-5}$, $t(4) = 15.1$) and KH ($p = 0.04$, $t(4) = 2.29$) media (Figs. 1G and 1H).

The autotrophic and heterotrophic growth rates of the wild-type and $M^-_3$ZFeL strains were evaluated using KH and CM culture media (Fig. 2). The mutant strain showed slightly slower and faster growth than did the wild type in the KH and CM culture, respectively. The $M_3^-$ZFeL culture ODs at the third ($p = 0.020$, $t(4) = -5.81$) and fourth ($p = 1.5 \times$

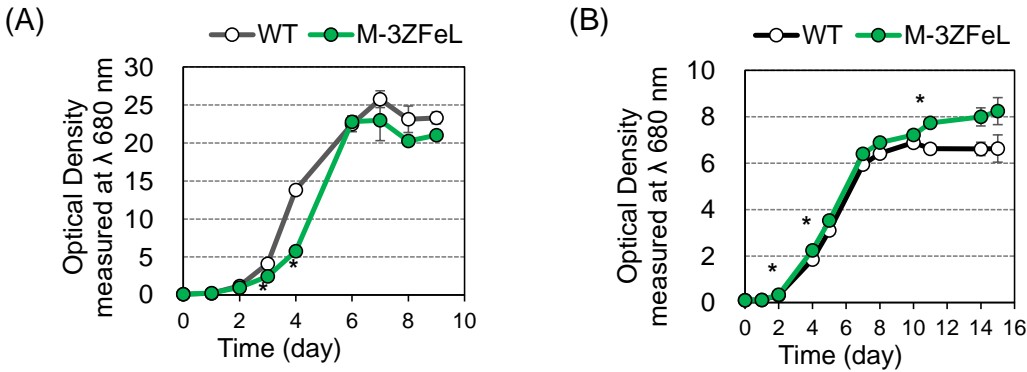

**Figure 2  Growth curves of M$_3^-$ZFeL and wild-type strains of *Euglena gracilis*.** (A and B) Growth curves of the wild-type (white) and M$_3^-$ZFeL (green) strains on heterotrophic KH (A) and autotrophic CM (B) media. The initial culture was inoculated into a medium with an optical density of 0.1 measured at λ 680 nm. The growth was determined on days 0, 1, 2, 3, 4, 6, 7, 8, and 9 after the start of the Koren–Hunter (KH) culture, while it was determined on days 0, 1, 2, 4, 5, 7, 8, 10 ,11, 14, and 15 after the start of the Cramer–Myers (CM) culture. The growth test was conducted in triplicate using three independent tubes. The error bars indicate the standard errors for these triplicates. *$p < 0.05$, Student's $t$-test with Bonferroni's correction.

$10^{-4}$, $t(4) = -20.48$) days of KH culture, which included high amounts of glucose, were significantly lower than those of the wild type (Fig. 2A). On the other hand, the ODs on the second ($p = 0.020$, $t(4) = -6.13$), fourth ($p = 0.042$, $t(4) = -4.98$), and eleventh ($p = 0.036$, $t(4) = -5.18$) days of CM culture were significantly higher than those of the wild type (Fig. 2B).

Our quantification of motility showed that the M$_3^-$ZFeL TM values were less than one-thousandth that of the wild type. Following previous studies, cells cultured in a liquid KH medium were adequately stirred and subjected to TM measurements. Figure 3A shows the cell distribution and swimming traces over approximately 8 s. The absence of swimming traces for the mutant strain revealed that cells could not move using their flagella. The total TM value ± SD for the wild-type strain was 11,161 ± 269 for 368 cells, whereas that for M$_3^-$ZFeL was only 9 ± 6 for 467 cells, as shown in Fig. 3B. The TM value for the mutant strain cannot be distinguished from the measurement noise, indicating that the M$_3^-$ZFeL strain was almost completely non-motile. Although some of the M$_3^-$ZFeL cells possessed a short flagellum (Figs. 1C–1H), our results indicated that the short flagellum did not function well.

## Potential of M$_3^-$ ZFeL for industrial application

M$_3^-$ZFeL demonstrated faster sedimentation than did the wild type in 1.5 mL microtubes. As shown in Fig. 4, the M$_3^-$ZFeL cell sedimentation rate was higher than that of the wild-type cells under both heterotrophic (Fig. 4A) and autotrophic (Fig. 4B) growth conditions using KH and CM media, respectively. The values at the final time point (40 min) were significantly different under both heterotrophic ($p = 0.010$, $t(4) = -3.69$) and autotrophic ($p = 0.011$, $t(4) = -3.60$) conditions (Figs. 4A and 4B).

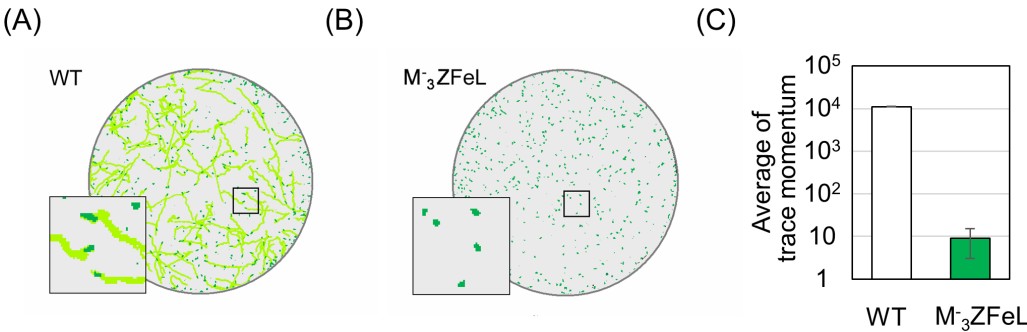

**Figure 3 Motility quantification of M$_3^-$ZFeL and wild-type strains of *Euglena gracilis* using trace momentum (TM).** (A–B) Cell distribution and swimming traces over approximately 8 s for the wild-type (A) and M$_3^-$ZFeL (B) strains. Both cell types were grown in heterotrophic (KH) medium and measured during the logarithmic growth phase. The magnified images of the areas within the squares are shown in the lower left of the panels. The diameter of these circles are approximately 2.5 mm. Green dots and light green lines indicate *E. gracilis* cells and their traces, respectively. (B) The motility of the wild-type and M$_3^-$ZFeL strains, evaluated using TM. The TM was obtained every 1.6 s and averaged over 10 min. Error bars show the standard deviation for 500 time points.

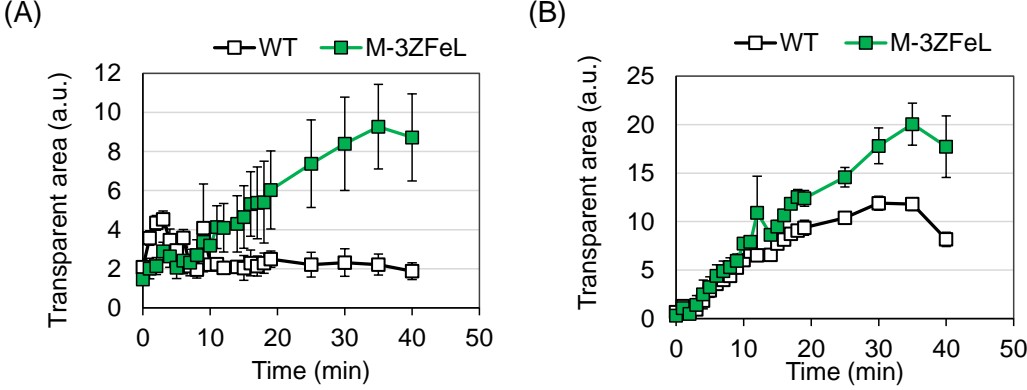

**Figure 4 Sedimentation speed of M$_3^-$ZFeL and wild-type strains of *Euglena gracilis*.** (A and B) Sedimentation rates of the wild type (white) and M$_3^-$ZFeL (green) in Koren–Hunter (KH) (heterotrophic; A) and Cramer–Myers (CM) (autotrophic; B) media. Photographs of the cell suspension were taken every 1 min to measure the transparent area in each 1.5 mL tube. After cell sedimentation, the supernatant becomes transparent, and the area on the photograph was used to evaluate the sedimentation rate. Error bars show standard errors for three replicates. a.u., arbitrary unit.

By analyzing sedimentation in a larger-scale culture, M$_3^-$ZFeL was found to be advantageous for industrial harvesting. To assess the effect of fast sedimentation on the harvest of cells cultured using photosynthesis, wild-type and M$_3^-$ZFeL strains were cultured using CM medium and their sedimentation rates were measured in 20 cm-deep beakers. M$_3^-$ZFeL cells showed a significantly faster sedimentation rate than did the wild-type cells (Fig. 5). The M$_3^-$ZFeL cell density at the bottom of the beaker (34.1 g L$^{-1}$), was more than 10 times that of the wild type (3.0 g L$^{-1}$).

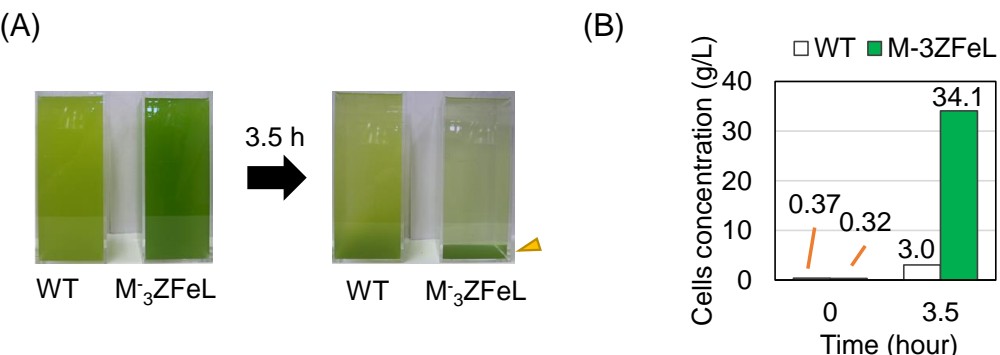

**Figure 5  Sedimentation rate of *Euglena gracilis* cells in 20 cm-deep beakers.** (A) Autotrophic cultures of the wild-type (left) and $M_3^-$ZFeL (right) strains suspended in 2 L beakers and left to stand under static conditions for 3.5 h. Cramer–Myers (CM) medium was used for the culture. (B) The sediment before and after 3.5 h of sedimentation. The cells that were retrieved from the bottom of the beaker and quantified for their dry weight are indicated with an arrowhead in (A).

The paramylon storage in $M_3^-$ZFeL cells was equal to or greater than that in the wild-type cells, whereas lipid accumulation in $M_3^-$ZFeL cells was less than (autotrophic) or equal to (heterotrophic) that that in the wild-type cells. As shown in Figs. 6A and 6B, $M_3^-$ZFeL cells produced a higher amount of paramylon (1.6 times; $p = 0.033$, $t(4) = -2.51$) than did the wild-type cells under autotrophic conditions, whereas there was no significant difference under heterotrophic conditions. In contrast, the $M_3^-$ZFeL strain showed a lower lipid content than did the wild type, especially under autotrophic conditions ($p = 0.034$, $t(4) = -2.47$). (Figs. 6C and 6D).

## DISCUSSION

The non-motile and fast sedimentation phenotypes of the $M_3^-$ZFeL mutant strain were considered to have strong correlations with the loss of negative gravitaxis. Wild-type *E. gracilis* exhibited positive and negative gravitaxis that depended on its environmental and cellular status (*Stallwitz & Hader, 1994*; *Lebert et al., 1999*). Recent research has shown that wild-type *E. gracilis* uses mechano-sensing proteins and flagellar beating to stay at a preferential water depth (*Häder & Hemmersbach, 2017*), rather than using buoyancy control. Since $M_3^-$ZFeL cells were incapable of flagellar beating, they could not use negative gravitaxis to prevent sinking. In addition, the highly accumulated paramylon, which had a particle density of approximately 1.5 in $M_3^-$ZFeL cells, may also contribute to the increase in their specific gravity and sinking under autotrophic conditions. Our sedimentation tests showed that the faster sedimentation of the $M_3^-$ZFeL cells was achieved not only in small microtubes but also inside a deeper reservoir, thus being advantageous for industrial harvesting. Typically, cells in heterotrophic cultures sediment faster than those in autotrophic cultures, probably due to the higher carbohydrate accumulation. However, our results in microtubes showed that both strains sedimented faster in the autotrophic culture than in the heterotrophic culture. This seems to be due to the differences in cell

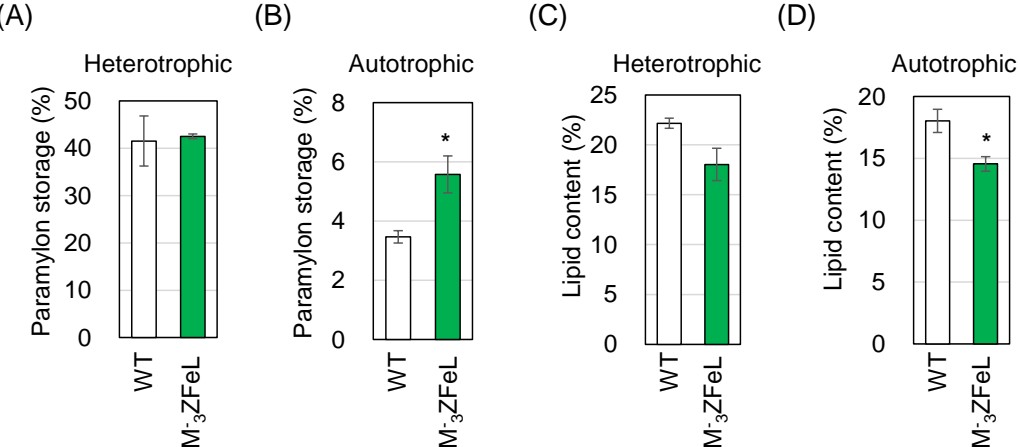

**Figure 6** **Paramylon storage and lipid content of M$_3^-$ZFeL and wild-type strains of *Euglena gracilis* cells.** (A–D) Paramylon storage (A and B) and lipid content (C and D) in the wild-type (white) and M$_3^-$ZFeL (green) strains cultured in heterotrophic (A and C) and autotrophic (B and D) media. Paramylon and lipid were extracted and quantified from freeze-dried cells, and their percentage weights were calculated. Cells were collected during their logarithmic growth phase. The y-axis scale is different in each figure. Error bars show standard errors for three replicates. *$p < 0.05$, Student's *t*-test.

density of the cultures; stacking highly concentrated cells at the bottom of the KH culture inhibited further sedimentation of the cells.

Flagellar motion is derived from the driving force of the doublet microtubules and dynein in the flagellum. Defects in the dynein, axonemal proteins, or related components result in a lack of flagellar movement (*Turner, 2006*; *Wang et al., 2014*). We have not yet identified the component, or more specifically, the gene, responsible for the immobility of the M$_3^-$ZFeL mutant strain. However, the lack of motility possibly reduces the cellular energy consumption and glycolysis. Such suppression is reported to result in enhanced starch accumulation in *Chlamydomonas* (*Hamilton, Nakamura & Roncari, 1992*; *Tunçay et al., 2013*), which may explain why greater paramylon storage was observed in the M$_3^-$ZFeL cells.

The M$_3^-$ZFeL strain showed slightly slower growth in the heterotrophic KH culture than did the wild type, suggesting that the function of glucose metabolism was slightly disturbed in the mutant. Meanwhile, the M$_3^-$ZFeL strain showed faster growth in the autotrophic CM culture, which did not include carbon sources, suggesting that the mutant strain had an intact photosynthesis system and consumed the photosynthesized resources slower than the wild type. Moreover, the M$_3^-$ZFeL had no defect in paramylon production, thus indicating that M$_3^-$ZFeL may improve industrial paramylon production. Although a limited defect was observed in the accumulation of lipids, M$_3^-$ZFeL would also be applicable to lipid production. The higher accumulation of paramylon in M$_3^-$ZFeL may be partially due to the energy conservation by not swimming. However, since the proliferation speed of M$_3^-$ZFeL was not drastically faster than the wild type, even in the autotrophic culture, the amount of energy conservation in M$_3^-$ZFeL does not seem significant.

The $M_3^-$ZFeL characterization results suggest that the mutant is defective in the components related to flagellum motion and/or formation that are not critical for survival and proliferation, at least under controlled laboratory conditions. The production of mutant strains by mutagenesis using high-LET irradiations, such as Ar-ion or Fe-ion irradiation, is advantageous because it causes a small number of large deletions in the genome with few side mutations (*Hirano et al., 2015*; *Kazama et al., 2017*). Although we could not identify the genetic cause of the phenotype in $M_3^-$ZFeL, the process of breeding these strains ensures minimal side mutations. This will enable the immediate industrial use of the strain by reducing the possibility of showing unexpected and undesirable traits, which may not be observed in the laboratory but appear under harsh and unstable outdoor-culture conditions.

The $M_3^-$ZFeL mutant of *E. gracilis* produced in this study is highly promising for the industrial production of food ingredients and chemical substances because it evidenced fast sedimentation, high paramylon storage, and a growth speed comparable to that of the wild type, at least under autotrophic conditions. In particular, the fast sedimentation of the mutant strain will save time and energy during the harvesting processes, which will contribute to the use of *E. gracilis* as a feedstock for biofuel by improving the balance between the energy invested to produce it and the energy output. In practical cultivation to produce paramylon and lipid under autotrophic conditions, processes to accumulate respective ingredients are added. In particular, cultured cells are subjected to nitrogen-restricted conditions to accumulate paramylon; the culture is then condensed and hypoxically conditioned to ferment the paramylon to wax ester (*Suzuki, 2017*). Our results showed that $M_3^-$ZFeL can competently accumulate larger amounts of paramylon under heterotrophic culture conditions than under autotrophic conditions; therefore, we suggest that they also show sufficient paramylon accumulation by nitrogen restriction (*Briand & Calvayrac, 1980*; *Sumida et al., 1987*), which enables the subsequent accumulation of wax ester (*Inui, Ishikawa & Tamoi, 2017*). In contrast, the fast-sinking trait of the cells may cause sedimentation in industrial-scale culture ponds during cultivation. To test this and improve the mass-cultivation method, further experiments using practical outdoor-culture ponds are required.

In addition to the production cost, non-motile cells have the advantage of being easy to observe and manipulate in a limited area under the microscope. This will be helpful for further basic studies on *E. gracilis*, which should include a long-lasting evaluation of a specific cell, e.g., observing its basic physiological phenomena, such as cell division (Fig. S1 and Supplemental Movie), as well as the micro-manipulation of single cells, such as that through single-cell electroporation, to deliver genes and proteins to the cells (*Ohmachi et al., 2016*).

## CONCLUSIONS

This study demonstrated the successful production of a motility-defective mutant of the microalga *E. gracilis*, which has been industrially used in recent years. The acquired mutant showed amoebic motility but no swimming using the flagellum, and accumulated more

carbohydrates than did the wild type under autotrophic culture conditions. However, the growth results indicated that the energy conservation due to the non-motile phenotype was not significant. Instead, owing to the inability to migrate and the accumulation of carbohydrates with high specific gravity, the cells sedimented faster during incubation. This trait indicates the potential to save energy for harvesting by introducing a natural sedimentation process. Therefore, the produced motility-defective mutant of *E. gracilis* and its derivatives may be useful for the industrial production of *E. gracilis*.

## ACKNOWLEDGEMENTS

The heavy-ion beam irradiation was conducted at the RI beam factory (RIBF), which is operated by the RIKEN Nishina Center and Center for Nuclear Study, University of Tokyo. The SEM images of the cells were obtained with the assistance of Dr. Toyooka (RIKEN, CSRS, JPN). We would like to thank Editage for English language editing.

### Funding

This work was funded by the ImPACT Program and SIP Program (Cabinet Office, Government of Japan). The funders had no role in study design, data collection and analysis, decision to publish, or preparation of the manuscript.

### Grant Disclosures

The following grant information was disclosed by the authors:
ImPACT Program and SIP Program (Cabinet Office, Government of Japan).

### Competing Interests

Shuki Muramatsu, Kohei Atsuji, Koji Yamada, Hideyuki Suzuki, Takuto Takeuchi, Yuka Hashimoto-Marukawa, Kengo Suzuki, and Osamu Iwata are employees of euglena Co., Ltd. which is a private company selling E. gracilis products. All other authors declare that they have no conflict of interest.

### Author Contributions

- Shuki Muramatsu, Kohei Atsuji and Kazunari Ozasa performed the experiments, analyzed the data, prepared figures and/or tables, authored or reviewed drafts of the paper, and approved the final draft.
- Koji Yamada conceived and designed the experiments, performed the experiments, analyzed the data, prepared figures and/or tables, authored or reviewed drafts of the paper, and approved the final draft.
- Hideyuki Suzuki, Takuto Takeuchi, Yuka Hashimoto-Marukawa and Yusuke Kazama performed the experiments, authored or reviewed drafts of the paper, and approved the final draft.
- Tomoko Abe, Kengo Suzuki and Osamu Iwata conceived and designed the experiments, authored or reviewed drafts of the paper, and approved the final draft.

## Data Availability

All of the raw data used in the study are available in the Supplemental Files.

## Supplemental Information

Supplemental information for this article can be found online at http://dx.doi.org/10.7717/peerj.10002#supplemental-information.

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
