# Peer review of "Isolation and characterization of a motility-defective mutant of Euglena gracilis"

_PeerJ, doi:10.7717/peerj.10002_

## Round 0.1 · original submission · Major Revisions

Editor comments
Major comment
1. The paper has to be revised for English by a fluent English speaker
2. Add references wherever needed in the text. Few places have been highlighted by the reviewers
3. Statistical analysis needs to be properly stated and reported. Mention the p-value wherever you talk about any significant differences in the result (e.g. L. 317).
4. In the introduction, please add a paragraph that includes previous attempts of producing locomotion defective mutants from other microalgae
5. Please do not mix M&M with results and do not mix results with discussion (e.g. L. 269, 272, L.295, L. 320-326). The text has to be fully revised (especially the first section in results). Do not repeat the results in the discussion.
6. Avoid any conclusions that you provide no experimental evidence for
7. Tune down the industrial applications to only “potential”.(L. 288) and probably it is better if you restrict it to the discussion section only
8. L.264-273. Remove discussion sentences. Also, the growth of both strains has to be statistically evaluated based on all the points not just selected ones. If you calculate the growth rates of both strains (using the stationary phase points), you would probably end up having very comparable numbers.

Minor comments
9. L. 46, state clearly your hypothesis. What you are stating now is an objective
10. L. 43, industrial applications such as what?
11. Add more references to the first paragraph in the introduction.
12. P. 84. Specific which culture collections are you referring to.
13. L. 87 replace based by “because of”
14. L. 86 change “is also exploited” to “has been exploited”
15. L.99 you do not have any evidence that the higher production of paramlon is because of energy conservation! So tune down the sentence “could be because of”
16. L.111 what is IAM?
17. L. 114, provide media composition or at least the differences between the two media “like carbon source”
18. L. 163, give the reason of why the OD was taken within seconds “before sedimentation occurs”
19. L. 358-360, the meaning is unclear, can you be more specific

·

Basic reporting

The work of Muramatsu et al. is a straightforward communication where Euglena gracilis cell suspensions were Fe-ion-irradiated and a mutant strain with defects in flagellum formation and locomotion was isolated and characterized. This paper was already subjected to two rounds of review in another journal, it is well written, contains the adequate background, and is well structured.

Experimental design

The research question is straightforward and the experiments made to characterize the obtained mutant are clear-cut.

Validity of the findings

The paper describes the isolation of a immotile Euglena gracilis mutant that may have potential in certain biotechnological applications.

Additional comments

The manuscript may benefit if the authors make some minor corrections:
-line 62: "taxes" should read "taxis"
-line 89: add "industry" after "food"
-line 257 should read: "As judged by both light and electron microscopy, the M -3 ZFeL cells were more rounded than the wild type cells."
-line 380: change "the immotile cells have
merits in observing and manipulating cells in the limited area under the microscope." to "the paralyzed cells can be observed and manipulated in a limited area under the microscope."
-line 398: change "are" to "may be".
-Legend to Figure 1: it reads "(A and B) Photographs of wild type (A) and M -3 ZFeL (B) grown in KH liquid medium...." should read "(A and B) Photographs of wild type (A) and M -3 ZFeL (B) colonies grown in KH liquid medium...."
-I suggest to use the terms "non motile" or "paralyzed" instead of "immotile".

·

Basic reporting

The authors present the results of the isolation and characterization of a novel low-motility strain of Euglena gracilis. They test several physiological parameters on it, stating that it has potential use for industrial applications. In general terms is a quite interesting study and the authors “created” the strain and show its potential uses and advantages. This research can have a high applicability and should be published.

Experimental design

The experimental design is appropriate for the study. Is recommended to include some statistical analysis that will consolidate the results of the study.

Validity of the findings

Novel findings that show the potential of "creating" ad-hoc algal strains for industrial use.

Additional comments

Really nice work. However, there are minor changes required. There is overlapping between the materials & methods and results sections, with part of the methodology being described in the results, which makes the paper a bit hard to follow. It is recommended to clearly separate both sections, leaving all the methodological work and description of protocols in the material and methods part.

Reviewer 3 ·

Basic reporting

The text is clear but needs professional English improvement.
The article is conforming to professional expression standards.
The work is simple in methodology, reach an important goal of made a relationship between traits and biocontrol performance, with a good approach to field conditions.
Literature references are sufficient in this field background and contextualized.
The article includes sufficient introduction and background to demonstrate how the work fits into the broader field of knowledge. Relevant prior literature is appropriately referenced.

Experimental design

The adopted methodology is correct and well described, with the necessary and sufficient details to be repeated. The experiments were well planned and met the assumptions of casual accuracy. Statistical analysis is correct and appropriated.

Validity of the findings

The results are conclusive and supported by the appropriate methodology, including statistical analysis. The hypotheses have been tested and proven.
Conclusions are well stated, linked to original research question & limited to supporting results.

Additional comments

The article is valuable, show in a very simple ways a traits characterization related to E. gracilis mutant strain from Fe-ion-irradiated cell suspensions, which shows defects in flagellum formation and locomotion. Thus, this study has great importance and applicability.
Nevertheless, text needs edition. Please revise the enclosed annotated manuscript.

Introduction
Lanes 80-81. Reference of this statement is missing.

Material and Methods
In general, make sure to leave a space between the temperature number and °C.
Lane 187. Define the light intensity used in this assay.
Lane 190. Please describe the equipment information used for taking the images.

Results
Lanes 223-228. This paragraph must be moved to M&M section. Please edit this paragraph as follows: “To produce immotile E. gracilis mutant strains, two rounds of mutant screening were conducted following Fe-ion beam irradiation at 50 Gy. About 1 × 105 mutagenized cells with independent genetic backgrounds were placed at one end of each culture dish filled with 10 mL of liquid KH medium with 0.5% methyl cellulose, which was added to solidify and prevent unexpected stirring of the culture. Culture dishes were illuminated from the opposite end with 50 μmol photons/m2s of fluorescent light.”
Lanes 233-242. This paragraph must be moved to M&M section. Please edit this paragraph as follows: “Cells were placed at the opening end of a 15 mL conical tube filled with 5 mL of liquid KH medium with 0.5% methyl cellulose, illuminated with 50 μmol photons/m2s of fluorescent light, and shaded with aluminum foil, except at the closing end. After 2 weeks of static horizontal incubation in the tubes, cells proliferated more than 1,000 times. Most of them were found at the unshaded end suspected to seek light. We collected about 1,000 of cells showing no movement from the initial inoculation position and cultured them for an additional 2 weeks. The proliferated cells were randomly isolated using FACS (MoFlo XDP, Beckman Coulter, Brea, CA) to establish clonal lines.”
Lane 245. You stated that only one genetically independent mutant strain was obtained from each population, but described two: M-3ZFeL and M-4ZFeL.

Lanes 320-322. Results do not support this statement.

Annotated reviews are not available for download in order to protect the identity of reviewers who chose to remain anonymous.
External reviews were received for this submission. These reviews were used by the Editor when they made their decision, and can be downloaded below.

---

## Round 0.2 · Minor Revisions

- L.60 add a reference after predators
- L62, add "respectively" after light
L. 119 replace quantified by measured
L. 285-287 this is a discussion
L. 292-297 Do you really think it is needed to mention the significant difference in these points. Does this change the conclusion?
L. 299 if this is your quantification why having the references (Ozasa et al.)
L304 why not giving the TM value plus minus
L. 345, 350, 368, 373 it is not common to refer to the figures in the discussion section! do you think this is necessary?

External reviews were received for this submission. These reviews were used by the Editor when they made their decision, and can be downloaded below.

---

## Round 0.3 · accepted · Accept

Thanks for taking care of all comments. The paper has improved significantly.

External reviews were received for this submission. These reviews were used by the Editor when they made their decision, and can be downloaded below.